# Fatigue Crack Initiation Change of Cast AZ91 Magnesium Alloy from Low to Very High Cycle Fatigue Region

**DOI:** 10.3390/ma14216245

**Published:** 2021-10-20

**Authors:** Stanislava Fintová, Libor Trško, Zdeněk Chlup, Filip Pastorek, Daniel Kajánek, Ludvík Kunz

**Affiliations:** 1Institute of Physics of Materials, Czech Academy of Sciences, Žižkova 22, 616 00 Brno, Czech Republic; chlup@ipm.cz (Z.C.); kunz@ipm.cz (L.K.); 2Research Centre, University of Žilina, Univerzitná 8215/1, 010 26 Žilina, Slovakia; libor.trsko@uniza.sk (L.T.); filip.pastorek@uniza.sk (F.P.); daniel.kajanek@uniza.sk (D.K.)

**Keywords:** Mg, fatigue mechanism, microstructure, focused ion beam (FIB), scanning electron microscopy (SEM)

## Abstract

Fatigue tests were performed on the AZ91 cast alloy to identify the mechanisms of the fatigue crack initiation. In different fatigue regions, different mechanisms were observed. In the low and high cycle fatigue regions, slip markings formation accompanied with Mg_17_Al_12_ particles cracking were observed. Slip markings act as the fatigue crack initiation sites. The size and number of slip markings decreased with decreased stress amplitude applied. When slip markings formation was suppressed due to low stress amplitude, particle cracking became more important and the cracks continued to grow through the particle/solid solution interface. The change of the fatigue crack initiation mechanisms led the S-N curve to shift to the higher number of cycles to the fracture, demonstrated by its stepwise character. A lower fatigue limit of 60 MPa was determined at 20 kHz for 2 × 10^9^ cycles compared to the 80 MPa determined at 60 Hz for 1 × 10^7^ cycles.

## 1. Introduction

Due to the high specific strength, low density, and good castability, Mg alloys are still widely used materials in many engineering applications in the automotive industry, aerospace, biomedicine, electrical engineering, etc. [1,2,3,4,5,6,7]. However, Mg alloys are also characteristic of poor formability and limited ductility at room temperature which is caused by the hexagonal close-packed (hcp) lattice. Plastic deformation at room temperature in hcp lattice is limited to the basal slip system and also the presence of Mg_17_Al_12_ intermetallic phase in magnesium alloys decreases their plastic deformation ability [8]. Another important aspect limiting the application of Mg alloys is their low corrosion resistance [9]. On the other hand, many of the Mg alloys are biocompatible, which means that they can be used as human body implants without any unwanted irritable reactions with human organisms [7,10,11,12,13,14,15]. In such a case, the high reactivity of the material can be considered as an advantage, because the dissolution of the implant after a certain time avoids re-operation for the implant removal (so-called biodegradation) [10,13,14,15].

The exact properties of Mg alloys are predetermined by their chemical composition and can be tailored by additional processing influencing material microstructure and phase distribution [1,2,3,4,5,6,9,16,17,18,19,20]. Magnesium alloys are, however, very often used in the as-cast state. The problem is, that casting leads to the creation of relatively large defects such as pores and shrinkages. These defects have a significant negative effect on resistance to cyclic loading and serve as fatigue crack initiation sites [6,21,22,23,24].

One of the most widely used Mg alloys is AZ91. The alloy exhibits a good combination of mechanical properties and corrosion resistance [1,2,9,25]. The content of Al increases material hardness and improves castability, while Zn in combination with Al also positively affects the alloy strength without reducing its ductility. The alloy, however, suffers from poor creep resistance at a temperature above 403 K, which was reported to be improved via alloying with different elements (Bi, Ca, Sb, Si, Pb, Sr, REs, etc.) [9]. These alloying elements were shown to be beneficial also in the microstructure modifications, suppressing Mg_17_Al_12_ phase formation [5], which has a direct impact on material mechanical properties and corrosion resistance. The alloy is also age hardenable and was shown to be suitable for mechanical processing and processing via severe plastic deformation, resulting in a significant mechanical and corrosion properties improvement [1,2,3,7,17,19,25,26,27,28,29,30]. Due to its properties, AZ91 is the preferential magnesium alloy for automotive and biomedical applications [7,31]. The automotive applications of AZ91 magnesium alloy cover transmission castings, instrument panels, cylinder heat cover, steering components, seat frames, oil pans, engine blocks, wheels, rims, induction clutch and brake pedals, miscellaneous components, etc. [31]. Although not all components are subjected to cyclic loading during their use, it is necessary to take into account at least their exposure to vibrations, which also causes cyclic loading and can reach a high number of cycles. Fatigue properties of AZ91 alloy are strongly limited by the low plasticity of the Mg hcp lattice and the presence of brittle Mg_17_Al_12_ phase. Fatigue data for as-cast AZ91 alloy are present in literature and most of the time the casting defects were shown to be responsible for the fatigue crack initiation and specimen failure [2,3,6,18,24]. However, when no defects are present in the cast microstructure, the slip markings (SMs), manifesting the localization of the cyclic plastic deformation formed in the solid solution areas, act as the stress concentrators for the fatigue crack initiation [3,17]. Their size and amount were shown to be determined by the applied stress amplitude and a reached number of cycles to the fracture. Nevertheless, these findings were reported for the low (LCF) and high cycle fatigue (HCF) regions but many of the applications were undergoing a higher number of cycles (very high cycle fatigue (VHCF) region). However, there is only very limited data available on the fatigue VHCF behavior of AZ91 in the literature [32]. Additionally, in this case, the casting defects were shown to be responsible for the fatigue crack initiation in the region exceeding 10^7^ cycles (over the HCF region) while probable fatigue crack initiation on the interface between Mg_17_Al_12_ phase particles and solid solution matrix was suggested by the authors too [32].

Since the only limited data about the fatigue crack initiation in defect-free AZ91 cast alloy are available in the literature, this paper is focused on the identification of the change of the fatigue crack initiation mechanism of cast AZ91 magnesium alloy from LCF to VHCF region. Polished specimens were subjected to the uniaxial fatigue loading in the tension-compression mode under low and very high frequencies and the resulting S-N curve has a stepwise character. The obtained results were discussed concerning the fatigue crack initiation mechanism.

## 2. Experimental Materials and Methods

### 2.1. Experimental Material

The chemical composition of the squeeze cast AZ91 alloy provided by the producer is shown in Table 1. The material was delivered in a form of bars with a diameter of 15 mm and a length of 80 mm.

Tescan LYRA 3 XMU FEG/SEM x FIB scanning electron microscope (SEM) (Tescan, Brno, Czech Republic) equipped with energy-dispersive X-ray spectroscopy (EDS, model X-Max 80, Oxford Instruments, Abingdon, UK) was used to analyze the experimental material microstructure. The analyzed specimens were conventionally mounted and metallographically prepared and etched by 2% Nital etchant. The microstructure of the examined cast AZ91 magnesium alloy is shown in Figure 1. The cast AZ91 alloy microstructure is formed by polyhedral grains of γ-phase (solid solution of alloying elements in magnesium), Mg_17_Al_12_ discontinuous precipitate (growing from the grain boundary towards the grain volume), eutectic (Mg_17_Al_12_ phase particles embedded in saturated γ–phase) and a high number of Mg_17_Al_12_ and AlMn based intermetallic particles which are more or less equally distributed in the volume of the microstructure. Observed phases were identified based on the EDS chemical analysis and literature. The average grain size of the γ–phase determined by the interception method on several areas using light optical microscope Olympus GX51 is 185 ± 70 µm.

Static mechanical properties of the cast alloy were determined via tensile and compression tests performed on a ZWICK Z020 testing machine (Zwick/Roell, Ulm, Germany) following the EN ISO 6892-1 standard [33]. For both tests, the test speed was 1 mm·min^−1^. Tensile tests were performed on cylindrical specimens with a gauge diameter of 6 mm and length of 30 mm. The ultimate tensile strength, σ_UTS_, proof stress obtained at 0.2 % of plastic deformation, σ_0.2_, and ductility were determined [17]. The strain was recorded with an extensometer. Compression tests were performed on cylindrical specimens with a diameter of 9 mm and a length of 14 mm. Both the compressive strength, σ_C_, and compressive proof stress obtained at 0.2 % of plastic deformation, σ_C0.2_ were determined. Elastic modulus, E = 44 GPa, of the alloy was determined using a frequency resonance technique performed on Grindosonic equipment according to the EN 843-2 standard [34].

As the proof stress in compression, σ_C_ = 317 ± 20 MPa, is almost the same as the one determined for the alloy in tension, σ_UTS_ = 167 ± 8 MPa [17], the symmetrical response of the material to the tension-compression cyclic loading is assumed. Similar values were determined also in the case of the proof stresses for compression (89 ± 8 MPa) and tension (87 ± 8 MPa, [17]).

### 2.2. Experimental Methods

To cover the wide range of fatigue lifetimes from the low to the very high cycle region, three types of testing machines were used. All the tests were performed under the load control mode in symmetrical tension-compression (load ratio R = −1).

Testing in the LCF and HCF region was performed on Shimadzu EHF-F1 servohydraulic system (Shimadzu, Duisburg, Germany. The geometry of the used specimens is shown in Figure 2a. The tests were performed at room temperature in laboratory air. The testing frequency was in the range from 0.1 to 10 Hz according to the cyclic response of the material on loading. As it was shown in [17], the cast material exhibits a short stage of cyclic softening, typically in the range of hundreds of cycles, followed by pronounced cyclic hardening characteristic for almost all remaining fatigue life. When the specimen lifetime exceeded 1 × 10^6^ cycles, the test was terminated and the specimen was moved to the Amsler HFP 5100 resonant fatigue testing device (Zwick/Roell, Ulm, Germany) and the experiment continued with the loading frequency of 60 Hz.

Fatigue tests in the VHCF region were conducted on a piezoelectric ultrasonic system made by Lasur^®^ (Lasur Sarl, Asnieres sur Seine, France). In ultrasonic testing, the displacement (i.e., the total strain amplitude) is controlled, however, due to negligible plastic strain amplitude it is equivalent to a load-controlled test. The tests were performed under the frequency of 20 kHz. The specimen for ultrasonic testing had to fulfil the resonance conditions [35]. The specimen geometry (cross-section and length of heads) was individually adjusted so the resonance frequency of the ultrasonic horn with the mounted specimen was in harmony with the resonance frequency of the ultrasonic horn. The geometry of the used specimens is shown in Figure 2b.

The fracture surface of the broken specimens was analyzed by SEM to identify the fatigue crack initiation site and fatigue crack initiation mechanism. The gauge length of the tested specimens was carefully mechanically polished by diamond paste (up to 1 μm) and slightly etched by 2% Nital before testing. Visualization of the microstructure allows SEM inspection of the surface to reveal the fatigue damage mechanism concerning the microstructural features. The roughness of the polished specimens measured using Mitutoyo SJ-201 P (Mitutoyo, Kawasaki, Japan) on 10 different specimens gauge lengths was Ra = 0.11 ± 0.03 μm.

Focused ion beam (FIB) cutting was used to reveal the microstructure below the specimen surface and to identify the fatigue crack initiation mechanism. The surface of the examined fatigue crack initiation site was protected by Pt layer and a FIB cut was performed.

## 3. Results and Discussion

### 3.1. Fatigue Tests

Fatigue test results obtained on the AZ91 cast magnesium alloy providing information about the alloy behavior in various fatigue regions are shown in Figure 3. The obtained S-N curve exhibits typical knee between the low (up to 10^5^) and high cycle (up to 10^7^) fatigue regions. Specimens considered as run-outs are marked by arrows in the graph. In the case of the HCF region, typically, the specimens exceeding 1 × 10^7^ cycles are considered as run-outs. In the case of the VHCF region, the run-out specimens have to reach 2 × 10^9^ cycles without failure to be considered as run-outs.

Since no specimen failure below the stress amplitude of 80 MPa up to 1 × 10^7^ cycles was observed, the value of 80 MPa was considered as the fatigue limit for the HCF region. Applying higher stress amplitude resulted in the failure of the tested specimens before 5 × 10^4^ cycles. The determined fatigue limit is, however, higher than the values presented in [6,24,36], where only 50 MPa as an endurance limit for AZ91D cast alloy was determined. The higher value of the fatigue endurance limit determined in the present work can be explained based on the microstructural analysis where alloy used in this work is typical by the absence of casting defects contrary to most of cast AZ91 Mg alloys reported in the literature. Authors in [6,24,36] observed casting defects in their materials, and these were responsible for the fatigue crack initiation in all the cases. However, the presented results are in good agreement with [2], where the average value of the endurance limit of 80–90 MPa was determined. The determined value of fatigue strength for conventional 1 × 10^7^ cycles is only slightly lower than the values of tensile and compression proof stress of the alloy. This only small difference between the cyclic and static characteristics of the material could be attributed to the cyclic hardening characteristic for almost the whole fatigue life of the alloy presented in [17].

Very high-frequency (20 kHz) loading was performed at almost identical stress amplitudes as the fatigue tests in the HCF region (tested at 10 + 60 Hz), when the specimens exceeded 1 × 10^7^ cycles and were considered as run-outs. An experiment performed at a stress amplitude of 60 MPa was terminated when the specimen lifetime exceeded 2 × 10^9^ cycles and the specimen was considered as a run-out. In [32] the fatigue strength for AZ91 cast alloys for 10^8^ cycles was estimated as 49.8 MPa, while the casting defects present in the microstructure were the most responsible factor for the fatigue crack initiation.

It is obvious from the results in Figure 3 that cycled AZ91 cast specimens break mostly in the LCF region. There is also an interesting plateau between 5 × 10^4^ and 5 × 10^6^ cycles where no specimen failed. This plateau could be explained by the changing mechanism of the fatigue crack initiation.

Multiple fatigue crack initiation was characteristic for AZ91 cast magnesium alloy for all the fatigue regions and all the applied stress amplitudes. The fracture surface was characteristic of the transcrystalline fatigue crack growth mechanism. Fatigue crack initiation places were localized on the specimen surfaces. No characteristic microstructural features, like casting defects, were observed serving as fatigue crack initiation places.

### 3.2. Fatigue Crack Initiation—Surface Analysis

Depending on the applied stress amplitude, the fatigue crack initiation mechanism has changed. In the fatigue region below 10^7^ cycles (LCF and HCF), fatigue crack initiation on SMs was observed (Figure 4a–c). In the VHCF region, the fatigue cracks initiation on the broken Mg_17_Al_12_ particles (Figure 5a,d) was the main mechanism responsible for the failure of the specimen. The intensity of SMs formation and evolution resulting in crack initiation was observed to be mainly dependent on the applied stress amplitude. In the case of specimens tested above the stress amplitude of 120 MPa, a large number of well-developed SMs localized in the solid solution areas was observed (Figure 4a). Additionally, a limited number of broken large intermetallic particles within the gauge lengths of examined specimens was observed (Figure 4b). Nevertheless, crack propagation from the particles to the solid solution was detected only rarely for these loading conditions.

Cyclic plastic deformation localization was manifested by SMs creation in the case of LCF and HCF regions. As a consequence of the created relief, the SMs acted as intense stress concentrators resulting in the fatigue cracks initiation sites. Since the majority of the energy for plastic deformation was released through the SMs formation, the cracks initiated in brittle intermetallic particles stopped propagating on the particle/solid solution interface.

A decrease in the stress amplitude below 120 MPa resulted in a decrease in SMs in both size and amount (Figure 4c). Even though also, in this case, the large intermetallic particles cracking was observed, the SMs were again responsible for the fatigue cracks initiation. In some cases, the growth of the crack from the broken particle to the solid solution was observed (Figure 4d). Fatigue cracks initiation at the surface micro-relief, at intrusion–extrusion interface, is in good agreement with [3,17].

In the case of tests performed at stress amplitudes close to the fatigue endurance limit of 80 MPa (determined for 1 × 10^7^ at 60 Hz), only a rare formation of SMs was observed. However, crack initiation on SMs was accompanied by the propagation of cracks formed in the intermetallic particles. This was not observed for higher testing stress amplitudes usually. Even though in this case, the cracks from the broken particles grew to the solid solution areas, cracks created on SMs were responsible for the failure of the tested specimens (Figure 4c,d). On the other hand, no SMs were observed on the specimen tested at 20 kHz, and the cracks propagating from the broken intermetallic particle were considered to be responsible for the specimen failure (Figure 5b,d). It could be attributed partially to (i) lower loading stress amplitudes and partially (ii) high test frequency as it was shown it has a great impact on the dislocation movement and internal lattice defects migration [37,38,39].

Reaching 10^7^ cycles was accompanied by an absolute change of the fatigue crack initiation mechanism. Below the stress amplitude of 80 MPa, any signs of the localization of cyclic plastic deformation manifesting in SMs formation were not observed. In the case of specimens reaching the number of cycles to the fracture above 10^7^ cycles, the cracks responsible for the specimen failure were identified to propagate from the broken intermetallic particles (Figure 5c,d). In the case of low used stress amplitudes (below 80 MPa) the response of the material to the cyclic loading did not lead to the formation of SMs because the applied stress amplitude was not sufficient to localize appropriate cyclic plastic deformation in the softer (more ductile) solid solution and the cracking of the brittle intermetallic particles became dominant. In some cases, also particle decohesion (particle/solid solution interface cracking) was observed. However, no relationship between the particle size and fatigue crack initiation was observed. Only some of the particles were broken while small particles were observed to be cracked as well as the large particles.

The observed change in the fatigue crack initiation mechanism is outlined in Figure 6. The obvious decrease in the SMs size and density can be attributed to the increase in the effective stress resulting in cross slip limitation and shortening the time for point defects migration described in [37,38,39]. As a result, the shape and density of SMs change, and mainly the kinetics of their formation slow down. As a result of the shift of the S-N curve to the higher number of cycles, an increase in the fatigue limit for the very high frequency (20 kHz) is usually observed. However, this is not the case. The results obtained at 20 kHz correspond to the data measured at 10 and 60 Hz. The decrease in the fatigue limit determined for 1 × 10^7^ and 2 × 10^9^ cycles is, therefore, a consequence of the change of the fatigue crack initiation mechanism. Since the SMs were not created below a stress amplitude of 80 MPa, the intermetallic phase particles with very different mechanical properties act as the stress concentrator, resulting in the fatigue crack initiation. As a result, the step-wise character of the S-N curve can be seen in Figure 3.

Since the decrease in the SMs creation was a consequence of the decrease in the loading (stress amplitude) and particle cracking was observed also for the HCF region, the influence of the frequency on the fatigue crack initiation can be considered as negligible for cast AZ91 alloy.

The two competing fatigue crack initiation mechanisms can explain the plateau existing between 5 × 10^4^ and 5 × 10^6^ cycles where no specimen failure occurred. While the introduced cyclic plastic strain was distributed more homogeneously in the material bulk—used for both the creation of the SMs and the intermetallic particles cracking, actually no specimen failure occurred. The fractured intermetallic particles consumed the available energy introduced by the presence of plastic strain necessary for the SMs creation and for the crack propagation into the more ductile solid solution. Thus the SMs creation and their participation in the specimen failure were suppressed. In the case of the particles cracking, no preferential grain orientation concerning the active slip system was necessary. Since the intermetallic particles were distributed in the microstructure more-less homogeneously, the localised plastic strain release was also distributed homogeneously. Due to a large number of small cracks on a large number of particles, a longer time (number of cycles) for the main crack to grow to the critical size to result in the specimen failure is needed compared to the case when only a limited number of cracks was created on the SMs, formed only under specific conditions due to the cyclic deformation localization in the hcp structure.

### 3.3. Fatigue Crack Initiation—FIB Cutting

While in the case of the LCF region the surface relief due to the SMs creation act as stress concentration sites and the fatigue cracks created on SMs were responsible for the specimen failure, in the case of VHCF and partially also in the case of HCF, the particles cracking and following crack propagation resulted in specimens failure. The different fatigue cracks initiation and following propagation in dependence on the applied stress amplitude and a reached number of cycles is documented in Figure 7 showing the FIB cuts performed on the tested specimens. The cuts were performed on broken specimens, however, at a sufficient distance from the fracture surface to eliminate the influence of the plastic zone formed at the crack tip during crack growth.

An example of a crack created on SMs, characteristic of the LCF region, is shown in Figure 7(a, detail A). In Figure 7(a, detail B) two cracks are documented, the crack propagating along the area of discontinuous precipitate and solid solution interface and broken particle. The FIB cut performed through the broken particle revealed that the crack did not continue to the solid solution matrix. Additionally, the second particle in the matrix, just below the specimen surface, can be seen. This intermetallic particle is broken as well, which indicates that the crack is not connected with the specimen surface, however, with the stress concentration within the material bulk.

On the other hand, the cracks propagating into the matrix can be seen in Figure 7b showing the subsurface microstructure of the specimen tested at 20 kHz reaching the VHCF region revealed by FIB cut. Even though no crack propagation into the matrix was observed on the surface, the cracks are growing into the specimen’s bulk. The cracks propagating into the material will grow and after reaching a certain size they will be revealed on the surface as documented in Figure 5c,d.

Both of the examples provide the proof that the brittle intermetallic particles act as the stress concentrators and act as the fatigue crack initiation site in the case of low used stress amplitudes. The difference is in the response of the more ductile matrix which can absorb some amount of the localized plastic deformation without failure (HCF region), however, reaching some threshold value results in crack propagation through the particle/solid solution interface (VHCF region).

## 4. Conclusions

Fatigue tests of AZ91 magnesium alloy from low to very high cycle fatigue regions were performed to describe the fatigue crack initiation mechanism and revealed its change. Based on the obtained results the following conclusions can be drawn:
Low and high (below 10^7^ cycles) cycle fatigue regions were characteristic by the fatigue crack initiation on the SMs created in the solid solution areas of the cast microstructure of AZ91 magnesium alloy. Additionally, broken Mg_17_Al_12_ intermetallic particles were observed in the microstructure, however, the cracks almost always stopped on the particle/solid solution interface and did not propagate to the material bulk.Fatigue cracks in the very high cycle fatigue region initiated on the broken primary Mg_17_Al_12_ intermetallic particles grow towards the solid solution areas.The change of the fatigue crack initiation mechanism led to the fatigue limit decrease from the value of 80 MPa determined from low-frequency tests data for 1 × 10^7^ cycles to 60 MPa determined from very high-frequency tests for 2 × 10^9^ cycles.


## Figures and Tables

**Figure 1 materials-14-06245-f001:**
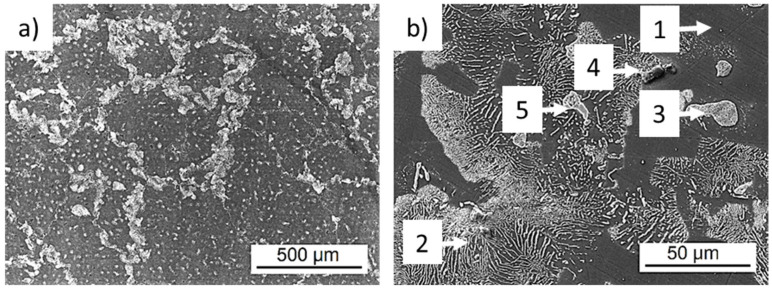
Microstructure of cast AZ91 magnesium alloy, etch; 2% Nital, SEM; (**a**) macro-view of the cast microstructure; (**b**) detail of microstructural features. 1—γ–phase, 2—Mg_17_Al_12_ discontinuous precipitate, 3—eutectic, 4—AlMn based intermetallic phase particle, 5—Mg_17_Al_12_ intermetallic phase particle.

**Figure 2 materials-14-06245-f002:**
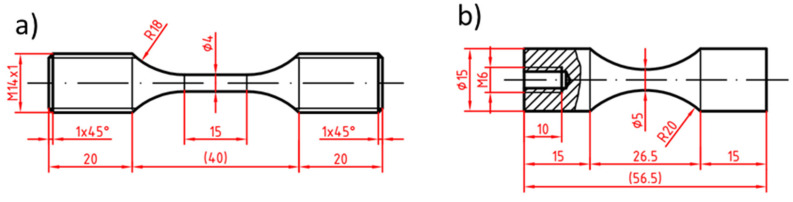
The geometry of fatigue tests specimens; (**a**) low and high cycle fatigue tests; (**b**) very high cycle fatigue VHCF tests. Dimensions are in mm.

**Figure 3 materials-14-06245-f003:**
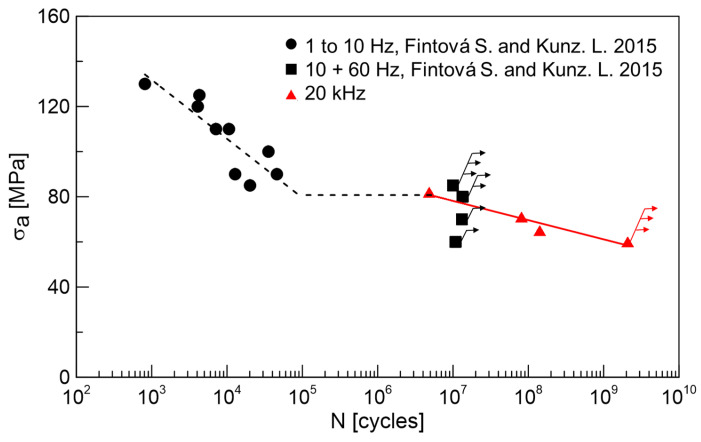
S-N curves for cast AZ91 magnesium alloy. Arrows indicate the run-out specimens, exceeding 1 × 10^7^ or 2 × 10^9^ cycles depending on the test conditions. Data of the tests performed up to 60 Hz (black symbols) were published by authors in [17].

**Figure 4 materials-14-06245-f004:**
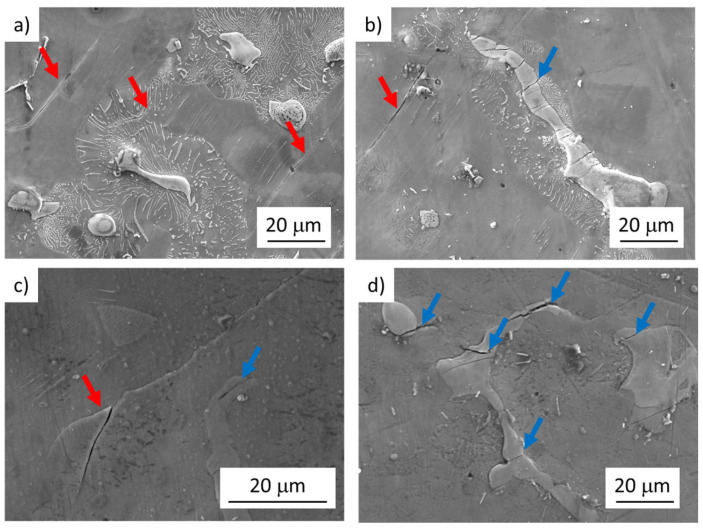
Fatigue crack initiation of AZ91 cast specimens tested at low frequencies. Red arrows indicate the cracks created on SMs. Blue arrows indicate the cracks in Mg_17_Al_12_ particles; (**a**) well developed SMs in the solid solution areas, and cracks on SMs; σ_a_ = 120 MPa, N_f_ = 4088 cycles, f = 10 Hz; (**b**) cracks on SMs and broken intermetallic particle; σ_a_ = 120 MPa, N_f_ = 4088 cycles, f = 10 Hz; (**c**) tiny SMs in the solid solution areas, cracks on SMs and broken intermetallic particle; σ_a_ = 85 MPa, N_f_ = 20,170 cycles, f = 60 Hz; (**d**) broken intermetallic particles with cracks growing to the material; σ_a_ = 85 MPa, N_f_ = 20,170 cycles, f = 60 Hz.

**Figure 5 materials-14-06245-f005:**
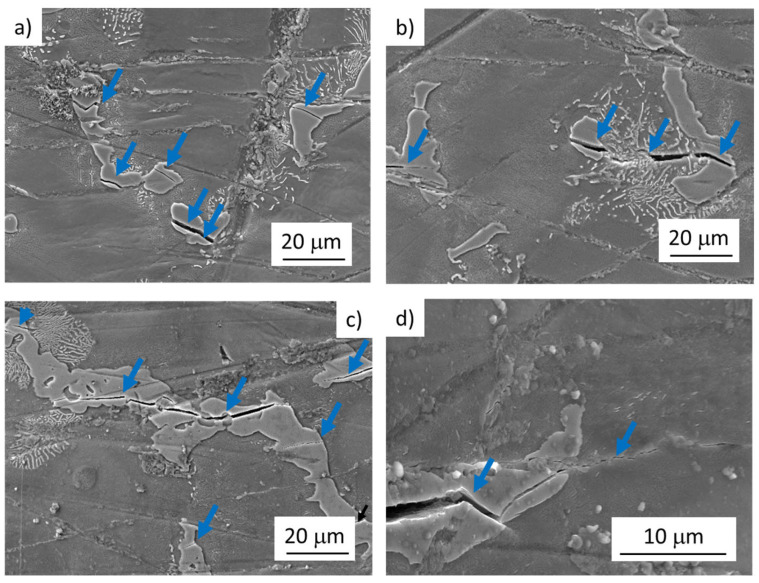
Fatigue crack initiation of AZ91 cast specimens tested at very high frequency. Blue arrows indicate the cracks in Mg_17_Al_12_ particles; (**a**) broken intermetallic particles; σ_a_ = 82 MPa, N_f_ = 4,841,760 cycles, f = 20 kHz; (**b**) broken intermetallic particles with cracks growing to the material; σ_a_ = 82 MPa, N_f_ = 4,841,760 cycles, f = 20 kHz; (**c**) broken intermetallic particles; σ_a_ = 65 MPa, N_f_ = 141,804,000 cycles, f = 20 kHz; (**d**) broken intermetallic particle with crack growing to the material; σ_a_ = 65 MPa, N_f_ = 141,804,000 cycles, f = 20 kHz.

**Figure 6 materials-14-06245-f006:**
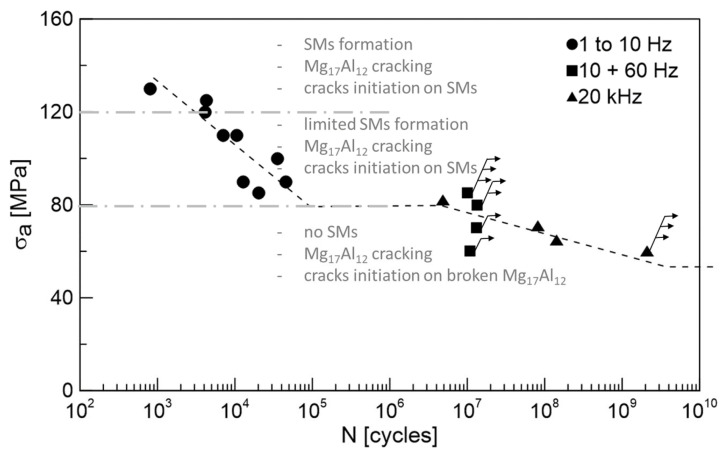
Change in the fatigue crack initiation mechanism for cast AZ91 magnesium alloy. Arrows indicate the run-out specimens, exceeding 1 × 10^7^ or 2 × 10^9^ cycles depending on the test conditions.

**Figure 7 materials-14-06245-f007:**
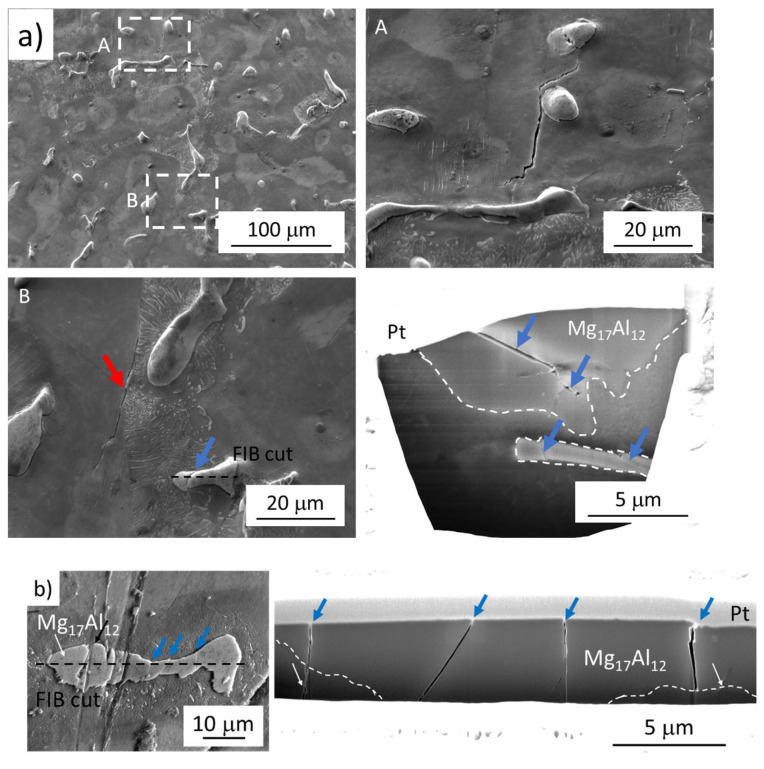
FIB cuts through the Mg_17_Al_12_ particles revealing the fatigue crack propagation mechanism. Blue arrows indicate cracks in the particles. Red arrows indicate the cracks in the solid solution matrix. The Mg_17_Al_12_ particle is outlined by white dashed line on FIB cuts; (**a**) LCF region; σ_a_ = 120 MPa, N_f_ = 4088 cycles, f = 10 Hz; (**b**) VHCF region; σ_a_ = 65 MPa, N_f_ = 141,804,000 cycles, f = 20 kHz.

**Table 1 materials-14-06245-t001:** Chemical composition of AZ91 magnesium alloy (in wt.%).

Elements	Al	Zn	Mn	Si	Fe	Be	Ni	Cu	Mg
wt.%	8.7	0.65	0.25	0.006	0.003	0.0008	0.0006	0.0005	balance

## Data Availability

The datasets generated during and/or analyzed during the current study are available from the corresponding author on reasonable request.

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
