# Peer review of "Fatigue Crack Initiation Change of Cast AZ91 Magnesium Alloy from Low to Very High Cycle Fatigue Region"

_materials, 2021, doi:10.3390/ma14216245_

Round 1

Reviewer 1 Report

Notes on the article of Stanislava Fintová, Libor Trško, Zdeněk Chlup, Filip Pastorek, Daniel Kajánek and Ludvík Kunz “Fatigue crack initiation change of cast AZ91 magnesium alloy from low to very high cycle fatigue region”

The paper reports results of studying the mechanisms of the fatigue crack initiation in the as cast alloy AZ91. The authors studied the fatigue life of the alloy in the wide range of number of cycles (until 2×109 cycles). The low (LCF), high (HCF) and very high cycle fatigue (VHCF) regions were investigated. The authors have done good research and analyzed the results carefully. The results of this article have the high theoretical and practical importance. This is an interesting and well-written report, which should be published after small revisions that are listed below:

  • Is there a relationship between the particle size of the phase and the rate of crack propagation?
  • It would be nice to measure the surface roughness of the fatigue test samples for easy comparison of different results in the future.

Author Response

Dear reviewer, thank you for your comments and questions.

Even though in the case of the low a middle frequency testing in some cases the particles failure was observed, the cracks mostly stopped on the particle/matrix boundary. The particles acted as “growing” fatigue crack initiation sites mostly in the case of the high-frequency testing. However, we did not measure the crack growth rate. Also, since not all of the particles were broken and served as crack initiation sites, it is difficult to determine which particle will be initiating the crack and analyze it. We did not observe any relationship between the particle size and crack initiation – large, as well as small particles were observed to be cracking.

The information about the surface roughness was added to the manuscript.

On behalf of authors

Stanislava Fintova

Reviewer 2 Report

Overall, the manuscript was well prepared, and lots of interesting results were reported. Before the acceptance, a minor revision is needed based on the following comments.

  1. The introduction mentioned a lot about the corrosion resistance of Mg alloys. Is it related to the main contents or focus of this paper?
  2. How did you determine the Mg17Al12, AlMn, and grain size based on Fig. 1?
  3. How did you measure the strain during tensile test?
  4. The white and black arrows indicated in Figs. 4 and 5 are difficult to be identified. It is suggested to change the colors.

Author Response

Dear reviewer, thank you for your comments and questions.

1. The introduction mentioned a lot about the corrosion resistance of Mg alloys. Is it related to the main contents or focus of this paper?

A: Actually, corrosion is mentioned always in combination with other properties (including mechanical properties) and in relation to the microstructure, which is predetermining all the material characteristics. We believe we did not mention corrosion resistance and ways of its improvement more than other properties, however, it is one of the main characteristics of the Mg alloys limiting their usage and it is important to be mentioned.

Anyway, we had tried to modify the text slightly.

2. How did you determine the Mg17Al12, AlMn, and grain size based on Fig. 1?

A: The particles and grain size were not determined based on Fig. 1, it is just illustrating the character of the microstructure. As the chemical composition and structure of the alloy is already well known, intermetallic phase particles' chemical composition was estimated based on EDS chemical analysis and literature. The grain size was determined based on image analysis of several micrographs of the microstructure obtained by light optical microscope as the conventional method.

The information was emphasized in the text.

3. How did you measure the strain during tensile test?

The strain was measured using an extensometer. The information was added to the text.

4. The white and black arrows indicated in Figs. 4 and 5 are difficult to be identified. It is suggested to change the colors.

The color of the white and black arrows was modified into red and blue, respectively.

Stanislava Fintova

Reviewer 3 Report

In this paper, for AZ91, a type of Mg alloy, the differences in the crack initiation and propagation mechanisms between LCF and HCF regions and VHCF regions were investigated through fatigue tests and morphological analyses using SEM. The difference in test results and the reasons for it are properly and logically presented, so it seems to have academic value. In order to improve the quality of the paper, however, I would suggest you consider the following opinions and correct several errors.

1.

In the introduction section, the need for this study to be conducted should be supplemented a little more. For example, it would be good to introduce the fields in which AZ91 is mainly used, especially the case that it is subjected to LCF, HCF and VHCF fatigue conditions.

2.

In line 32, “Mg alloys” instead of “alloys” will be more readable.

3.

The contents of AZ91, such as polyhedral grains of γ-phase, Mg17Al12 precipitate and the eutectic which are introduced in lines 88-92, should be displayed in Fig.1 as well.

4.

I don’t understand clearly what the sentence in lines 135-136, which is starting with “The gauge length ~”, means. And there seems a missed letter in the unit.

Author Response

A: Dear reviewer, thank you for your comments and questions.

  1. In the introduction section, the need for this study to be conducted should be supplemented a little more. For example, it would be good to introduce the fields in which AZ91 is mainly used, especially the case that it is subjected to LCF, HCF and VHCF fatigue conditions.

A: Even though the alloy is not primarily subjected to direct VHCF loading, all the components in the automotive are at least subjected to vibration, which can be, however, also considered as cyclic loading. The knowledge of the material behavior is therefore essential for its application to predetermine the component lifetime and safety.

Some applications were added to the manuscript and the importance of knowing the fatigue properties was highlighted.

  1. In line 32, “Mg alloys” instead of “alloys” will be more readable.

A: The text was modified.

  1. The contents of AZ91, such as polyhedral grains of γ-phase, Mg17Al12precipitate and the eutectic which are introduced in lines 88-92, should be displayed in Fig.1 as well.

A: The figure was modified.

  1. I don’t understand clearly what the sentence in lines 135-136, which is starting with “The gauge length ~”, means. And there seems a missed letter in the unit.

A: The unit was corrected, it should be micrometer.

Stanislava Fintova

Reviewer 4 Report

The paper discusses the mechanism of fatigue crack initiation in  AZ91 magnesium alloy  from low to very high cycle fatigue

The paper is well-structured and well-written. However, the following discussions might improve the quality of the manuscript:

-The last line of the abstract, the results are given in too details which is not compatible with the rest of the abstract. It is recommended to be revised to have a more consistent text.

-At the end of the last paragraph, partly the results is given which is not suitable in the introduction. In introduction, normally the structure of the paper (basically the description of the works not results) is given in the last paragraph.

- Which standard is taken for carrying out the static tests? The geometry of the specimens is not well-described (one can not get a good imagination based on cylindrical specimens with a gauge diemeter….)

-It is not mentioned why different geometries are used for LCF/HCF and VHCF.

-Dementias in Figure 2 is not mentioned that are in millimeter.

-In line 126, something is missing (up to 1  m????)

-What is the definition of crack initiation in this paper in terms of length? Was it kept constant for all the specimens?

- How many tests were done in this work for each set of tests? The results shown in figure 3 are the results of the individual specimens or the average?

Author Response

Dear reviewer, thank you for your comments and questions.

-The last line of the abstract, the results are given in too details which is not compatible with the rest of the abstract. It is recommended to be revised to have a more consistent text.

A: Actually, we believe the abstract is consistent and also other reviewers were satisfied. The abstract should provide description of the problem, used methodology and results. It is not possible to describe the other results in more detail (change of the fatigue crack initiation sites from SMs to particles) and to describe the lifetime in a less detailed way (the numbers are the result).

-At the end of the last paragraph, partly the results is given which is not suitable in the introduction. In introduction, normally the structure of the paper (basically the description of the works not results) is given in the last paragraph.

A: The last sentence was replaced with a less concluding one.

- Which standard is taken for carrying out the static tests? The geometry of the specimens is not well-described (one can not get a good imagination based on cylindrical specimens with a gauge diemeter….)

A: The tensile tests were carried out according to the EN ISO 6892-1 standard. The information was added to the text.

-It is not mentioned why different geometries are used for LCF/HCF and VHCF.

A: The specimen had to fulfill the resonance conditions, which means the geometry had to be designed according to the used material and its acoustic properties. Whole the system (specimen mounted to the horn) has to be in harmony with the resonance frequency of the ultrasonic horn.

In the paper there is not sufficient space to explain whole the calculations, however, the relevant reference was added so the reader can find more information if interested.

  1. Bathias, C.; Paris, P.C. Gigacycle Fatigue in Mechanical Practice; Taylor & Francis: 2004.

-Dementias in Figure 2 is not mentioned that are in millimeter.

A: The information was added to the figure description.

-In line 126, something is missing (up to 1  m????)

A: The text was corrected – unit should be micrometer.

-What is the definition of crack initiation in this paper in terms of length? Was it kept constant for all the specimens?

A: The fatigue crack was random. In our case, the fatigue life was the main topic of the paper and the cracks we were able to observe were secondary cracks, not the main crack responsible for specimen failure. However, after analysis of the main crack, also the secondary cracks were analyzed, exhibiting the same character (in terms of the crack initiation sites). So we provided the images of the secondary cracks and results of the mechanism analysis.

- How many tests were done in this work for each set of tests? The results shown in figure 3 are the results of the individual specimens or the average?

A: Each point in the graph represents one test. As you can see, the tests were performed also at the frequency of 1 Hz, which is quite time-consuming. However, to get some statistics, several specimens were tested at the same stress amplitude. Taking into account the fact, that a cast alloy (always exhibiting some range of results) was tested, we believe, that sufficient tests were performed. Our point was to analyze the fatigue crack initiation mechanism and to describe it, eventually, reveal the differences in individual fatigue regions. Since the same fatigue crack initiation character was observed for the specimens tested in the same fatigue region, there was no need to repeat the tests.

Stanislava Fintova